# A simplified, combined protocol versus standard treatment for acute malnutrition in children 6–59 months (ComPAS trial): A cluster-randomized controlled non-inferiority trial in Kenya and South Sudan

Jeanette Bailey[1,2]*, Charles Opondo[3], Natasha Lelijveld[4], Bethany Marron[1], Pamela Onyo[5], Eunice N. Musyoki[6], Susan W. Adongo[6], Mark Manary[7], André Briend[8,9], Marko Kerac[2,10]

1 International Rescue Committee, New York, New York, United States of America, 2 Department of Population Health, London School of Hygiene & Tropical Medicine, London, United Kingdom, 3 Department of Medical Statistics, London School of Hygiene & Tropical Medicine, London, United Kingdom, 4 No Wasted Lives, Action Against Hunger UK, London, United Kingdom, 5 Action Against Hunger, Juba, South Sudan, 6 International Rescue Committee, Nairobi, Kenya, 7 Department of Pediatrics, Washington University School of Medicine, Saint Louis, Missouri, United States of America, 8 Department of International Health, University of Tampere, Tampere, Finland, 9 Department of Nutrition, Exercise and Sports, University of Copenhagen, Copenhagen, Denmark, 10 Centre for Maternal, Adolescent, Reproductive, & Child Health, London School of Hygiene & Tropical Medicine, London, United Kingdom

* jeanettebailey1@gmail.com

**Data Availability Statement:** Underlying data, code and supporting documentation for this paper

## Abstract

### Background

Malnutrition underlies 3 million child deaths worldwide. Current treatments differentiate severe acute malnutrition (SAM) from moderate acute malnutrition (MAM) with different products and programs. This differentiation is complex and costly. The Combined Protocol for Acute Malnutrition Study (ComPAS) assessed the effectiveness of a simplified, unified SAM/MAM protocol for children aged 6–59 months. Eliminating the need for separate products and protocols could improve the impact of programs by treating children more easily and cost-effectively, reaching more children globally.

### Methods and findings

A cluster-randomized non-inferiority trial compared a combined protocol against standard care in Kenya and South Sudan. Randomization was stratified by country. Combined protocol clinics treated children using 2 sachets of ready-to-use therapeutic food (RUTF) per day for those with mid-upper arm circumference (MUAC) < 11.5 cm and/or edema, and 1 sachet of RUTF per day for those with MUAC 11.5 to <12.5 cm. Standard care clinics treated SAM with weight-based RUTF rations, and MAM with ready-to-use supplementary food (RUSF). The primary outcome was nutritional recovery. Secondary outcomes included cost-effectiveness, coverage, defaulting, death, length of stay, and average daily weight and MUAC

are available at https://datacompass.lshtm.ac.uk/1151/. The full dataset cannot be made open access due to the presence of participant identifiable content. However, a redacted version will be provided to interested parties, subject to the completion of a request form (available via the repository link above) and signing of a Data Transfer Agreement.

**Funding:** This study was funded by the United States Agency for International Development Office of U.S. Foreign Disaster Assistance (https://www.usaid.gov/who-we-are/organization/bureaus/bureau-democracy-conflict-and-humanitarian-assistance/office-us), grant number AID-OFDA-G-14-00208 awarded to the International Rescue Committee, and the Children's Investment Fund Foundation (https://ciff.org/) grant reference title 'Reinventing community management of acute malnutrition' awarded to Action Against Hunger. The funders had no role in study design, data collection and analysis, decision to publish, or preparation of the manuscript.

**Competing interests:** The authors have declared that no competing interests exist.

**Abbreviations:** ComPAS, Combined Protocol for Acute Malnutrition Study; MAM, moderate acute malnutrition; MUAC, mid-upper arm circumference; NGO, nongovernmental organization; OTP, outpatient therapeutic program; RUSF, ready-to-use supplementary food; RUTF, ready-to-use therapeutic food; SAM, severe acute malnutrition; SDG, Sustainable Development Goal; SFP, supplementary feeding program; SQUEAC, Semi-Quantitative Evaluation of Access and Coverage; WHZ, weight-for-height $z$-score.

gains. Main analyses were per-protocol, with intention-to-treat analyses also conducted. The non-inferiority margin was 10%. From 8 May 2017 to 31 March 2018, 2,071 children were enrolled in 12 combined protocol clinics (mean age 17.4 months, 41% male), and 2,039 in 12 standard care clinics (mean age 16.7 months, 41% male). In total, 1,286 (62.1%) and 1,202 (59.0%), respectively, completed treatment; 981 (76.3%) on the combined protocol and 884 (73.5%) on the standard protocol recovered, yielding a risk difference of 0.03 (95% CI −0.05 to 0.10, $p = 0.52$; per-protocol analysis, adjusted for country, age, and sex). The amount of ready-to-use food (RUTF or RUSF) required for a child with SAM to reach full recovery was less in the combined protocol (122 versus 193 sachets), and the combined protocol cost US$123 less per child recovered (US$918 versus US$1,041). There were 23 (1.8%) deaths in the combined protocol arm and 21 (1.8%) deaths in the standard protocol arm (adjusted risk difference 95% CI −0.01 to 0.01, $p = 0.87$). There was no evidence of a difference between the protocols for any of the other secondary outcomes. Study limitations included contextual factors leading to defaulting, a combined multi-country power estimate, and operational constraints.

## Conclusions

Combined treatment for SAM and MAM is non-inferior to standard care. Further research should focus on operational implications, cost-effectiveness, and context (Asia versus Africa; emergency versus food-secure settings). This trial is complete and registered at ISRCTN (ISRCTN30393230).

## Trial registration

The trial is registered at ISRCTN, trial number ISRCTN30393230.

## Author summary

### Why was this study done?

- For decades, children with severe and moderate acute malnutrition have been treated separately, using different feeding products, protocols, and programs.

- Low coverage limits the global impact of acute malnutrition treatment programs.

- To improve impact and efficiency, we designed a simplified and combined approach to treat acute malnutrition, using mid-upper arm circumference to determine therapeutic food dosage.

- We hypothesized that this new approach would be as effective but more cost-effective than standard care, thereby raising the potential for improved future treatment availability.

## What did the researchers do and find?

- We conducted a cluster-randomized trial in Kenya and South Sudan to assess whether recovery in a simplified, combined protocol was non-inferior to standard treatment. Our main secondary objective was to assess cost-effectiveness.

- The needed sample size was calculated for a per-protocol analysis, with a target of 12 clusters of 100 children in each arm (2,400 children in total).

- Recovery in the combined protocol arm was non-inferior to that in the standard protocol arm, with 76.3% recovering in the combined protocol arm and 73.5% recovering in the standard protocol arm (risk difference of 0.03, 95% CI −0.05 to 0.10, $p = 0.52$).

- The average amount of ready-to-use food required for a child with severe malnutrition to reach full recovery was less in the combined protocol arm than in the standard protocol arm (122 versus 193 sachets), and the combined protocol was US$123 less per child recovered (US$918 versus US$1,041).

## What do these findings mean?

- A simplified protocol for severe and moderate acute malnutrition is as effective as standard treatment, achieving the same numbers of children recovered.

- The simplified, combined protocol saves money. Resource-constrained health systems could treat more children for the same investment, thereby extending coverage and improving public health impact.

- Prolonged and severe emergencies like the COVID-19 pandemic may increase global food insecurity and malnutrition and strain health systems further, increasing the need for simple, easy-to-use, and cost-effective malnutrition treatment approaches that reduce the strain on health workers and optimize health system resources.

## Introduction

Malnutrition is a major global public health problem, contributing to the deaths of approximately 3 million children under age 5 years each year [1]. Acute malnutrition affects more than 50 million children under age 5 and is particularly serious: As well as having a high short-term case fatality rate, it also has important long-term sequelae [2–5]. Tackling malnutrition is unfinished business from the Millennium Development Goals and remains prominent in the Sustainable Development Goals (SDGs). SDG 2.2 aims to "end all forms of malnutrition, including achieving by 2025 the internationally agreed targets on stunting and wasting in children under 5 years of age" [6]. Despite its importance, there is increasing concern that current approaches to acute malnutrition are suboptimal [7–9]. Successful treatments are available, but treatment program coverage is low, with at least 75% of acutely malnourished children 6–59 months old not accessing care [10,11]. This makes achieving impact at national and international levels difficult.

Acute malnutrition can be seen as a continuum condition, but current treatments for acute malnutrition are separated into 2 components: those treating severe acute malnutrition (SAM)

and those treating moderate acute malnutrition (MAM). Children with SAM have the highest risks for morbidity and mortality, but children with MAM remain at risk for adverse outcomes including illness and death [1,12–16]. SAM is currently defined as weight-for-height < −3 standard deviations (z-scores) below the WHO reference median and/or a mid-upper arm circumference (MUAC) < 11.5 cm and/or edema; MAM is currently defined as weight-for-height from −3 to <−2 standard deviations (z-scores) below the WHO reference median and/or a MUAC from 11.5 cm to <12.5 cm.

SAM and MAM are managed in separate programs, using different food products and protocols. There is currently no globally accepted guidance for the treatment of MAM, and MAM is not always routinely treated [17,18]. International mandate adds an additional layer of complexity: UNICEF supports the treatment of SAM and provides ready-to-use therapeutic food (RUTF) for use in outpatient therapeutic programs (OTPs); the World Food Programme supports the treatment of MAM and provides ready-to-use supplementary food (RUSF) or fortified blended flours for use in supplementary feeding programs (SFPs) [19,20]. In humanitarian settings, providing treatment for both SAM and MAM adds to the logistical and financial burden of health systems. When resources are scarce, and in the many settings where prevalence is not high enough to reach emergency thresholds, treatment of SAM is often prioritized, and children with MAM may not be eligible to receive care unless they deteriorate.

The Combined Protocol for Acute Malnutrition Study (ComPAS) unified the treatment of uncomplicated SAM and MAM for children 6–59 months into one protocol, with simplified diagnostic criteria and a single therapeutic food product. This is the first randomized controlled trial we are aware of to test a simple, MUAC-based dosage protocol for the treatment of SAM and MAM in children 6–59 months. A clinical trial assessing the effectiveness of an integrated SAM/MAM approach using a weight-based dosage protocol found similar recovery and lower therapeutic food costs in the integrated treatment arm [21]. An operational study of children admitted with a MUAC < 12.5 cm and/or edema and treated with a gradually reduced dosage of RUTF based on a combination of MUAC status and weight also found a high overall recovery rate [22]. Multiple studies assessing the use of MUAC in nutrition programs have shown that MUAC is effective at identifying children most at risk of mortality and is simpler and easier to use than weight-for-height measures [23,24]. MUAC gain can be used as a proxy for weight gain [25–28], and rate of weight and MUAC gain in response to treatment appears to decline as children recover [25,26,28,29]. Current standard of care provides a weight-based dosage of up to 200 kcal/kg/day for all children with SAM (S1 Table), but this dosage is based on the rate of weight gain in inpatient settings, which is higher than in outpatient settings [29]. Several recent studies of children with SAM in outpatient settings indicate that recovery with a reduced dosage is similar to recovery with standard treatment [21,22,30,31]. A study of children treated for SAM in Myanmar reduced the dosage to 500 kcal/day of RUTF once children reached a MUAC of 11.0 cm and a weight-for-height z-score (WHZ) ≥ −3 and achieved recovery above Sphere standards [30]. A study in Burkina Faso found that a reduced dosage of RUTF offered to SAM patients beginning in the third week of treatment resulted in non-inferior weight gain velocity and similar recovery and length of stay, though height gain velocity was reduced [31]. As children recover from SAM to MAM, current global practice reduces the dosage of energy received to provide a supplement rather than a replacement for the family diet, typically offering 1 sachet of RUSF per day (500–550 kcal/day) [18,32]. We previously conducted a secondary analysis of program data from 5 countries and found that children recovering from acute malnutrition with a MUAC < 12.5 cm may require 1,000 kcal per day, a reduced dosage compared to standard care [25]. The rationale for a reduced dosage is to facilitate increased coverage—and in turn increased public health impact—of treatment in a resource-constrained environment. The optimal dosage achieves the right

balance between meeting individual energy needs and extending treatment to more children. This study contributes to the evidence on the impact of different dosage regimes.

This study aimed to test the hypothesis that the combined protocol would be non-inferior to the standard protocol in terms of recovery, and improve cost-effectiveness.

## Methods

### Study design

ComPAS was a cluster-randomized non-inferiority trial conducted in 24 health facilities in Nairobi, Kenya, and Aweil East, South Sudan. We selected a cluster-randomized design in order to ensure fidelity to the assigned protocol at the health facility level, and to improve disaggregation of cost data by protocol type. We tested for non-inferiority, assuming recovery under the simplified protocol would be at least as good as under standard care. The clinical trial described in this paper builds on analysis conducted by our team to develop a simplified, combined protocol [25]. We previously published a full description of the protocol for this trial, as well as the methods for the cost-effectiveness analyses [33,34]. Ethical approval was given by the London School of Hygiene & Tropical Medicine (reference 11826), the Kenya Medical Research Institute (reference non-KEMRI 551), and the Ministry of Health in South Sudan (approved 21 November 2016).

### Participants

Aweil East, South Sudan, is a rural, agro-pastoralist region in Northern Bahr el Ghazal State with a total population of 309,921 and an under 5 population of 59,574 [35]. Nairobi, Kenya, is an urban area with a total population of 3,078,108 and an under 5 population of 462,849 [36]. Three sub-counties of Nairobi were included in the study: Embakasi North, Embakasi East, and Embakasi West.

Health facilities were eligible for selection as clusters. Only 12 health facilities were operating nongovernmental organization (NGO)–supported nutrition services in Aweil East at the time of randomization, and all were selected for participation. In Nairobi, the Ministry of Health aided selection of 12 health facilities in 3 sub-counties with the highest burden of malnutrition. Of the 32 health facilities in the 3 sub-counties of Nairobi, clinics were selected based on the following factors: the level of care provided (hospitals and dispensaries excluded), the type of care provided (routine child health services available), the population served (slum or peri-urban communities), and expected caseload of malnutrition. Children aged 6–59 months presenting to any of the 24 clinics involved in the study were eligible to enroll (S2 Table). Active case finding was also conducted by community health workers using MUAC to find and refer children. Active case finding was conducted in the same way for all clinics across both treatment arms and countries. Children admitted with a MUAC < 12.5 cm and/or edema (+/++, i.e., mild or moderate) were eligible for inclusion. Children exhibiting signs of a severe illness or danger sign according to the Integrated Management of Childhood Illness (IMCI) algorithm, or not passing the appetite test (consumption of 30 g of RUTF within 20 minutes), were excluded and referred for further assessment and care at an inpatient stabilization center or hospital [33,37].

Clinic-level and individual consent were obtained. Local health authorities and clinic managers gave written permission for their clinics to be included prior to randomization. For individual caretakers of malnourished children, participation was first explained in their local language (Dinka in South Sudan; Kiswahili or English in Nairobi). Caretakers were provided with an information sheet describing details of the interventions, and staff emphasized that treatment would be available to all regardless of whether they chose to participate. A consent

document was signed by caretakers, or by a witness who could attest to the caretaker's verbal consent.

## Randomization and blinding

The 24 health clinics were indexed and sent to the senior statistician (CO) based in London, who did not have any prior knowledge of their identities or characteristics. The statistician generated a randomized sequence list stratified by country using the Sealed Envelope randomized sequence generator and applied it to the list of the health facilities [38]. Participants were enrolled in the study arm of the clinic they presented to, by the research officer based at each clinic. Given that the intervention was applied at the facility level, clinic staff could not be blinded to group allocation, and neither could participants. To ensure objective trial management, the principal investigator (JB) was blinded to clinic allocation and outcomes by treatment arm until the last participant exited the trial and the database was locked. For data monitoring purposes during the trial, a data manager sent de-identified data to the PI, with site codes replacing clinic names and all information identifying clinics or treatment arms removed. The senior statistician revealed clinic allocation only after the trial database had been closed [33].

## Study procedures

In standard protocol clinics, children with SAM received RUTF according to weight (200 kcal/kg/day) in the OTP (S2 Table). Children with MAM received 500 kcal/day of RUSF (1 sachet/day) in the SFP. The OTPs and SFPs in the control arm in both countries operated from the same clinics with the same staff. In combined protocol clinics, children with a MUAC < 11.5 cm and/or edema (+/++) received 1,000 kcal RUTF/day (2 sachets/day), and children with a MUAC of 11.5 to <12.5 cm and no edema received 500 kcal RUTF/day (1 sachet/day). When children met the criteria for transition from SAM to MAM (MUAC ≥ 11.5 cm and no edema for 2 consecutive visits), they were switched from 2 sachets of RUTF/day to 1 sachet of RUTF/day. This reduction in dosage is based on the slowing rate of growth that we observed at a MUAC of approximately 11.5 cm in our stage 1 study [25] and aligns with UN guidance on the treatment of MAM [18,32]. Children in the combined protocol arm were registered as either an OTP or SFP patient based on their admission MUAC.

In the standard protocol, children were only included in this analysis if their MUAC on admission was <12.5 cm and/or they had edema (+/++). If they also met the WHZ < −2 criterion on admission and reached WHZ ≥ −2 prior to reaching MUAC ≥ 12.5 cm during treatment, they were not discharged until they had a MUAC measurement of ≥12.5 cm for 2 consecutive visits, to ensure parity of the definition of recovery between the intervention and control arms.

In both arms, children with SAM came once a week for a medical and nutritional evaluation and received systematic medications per the national protocol. Children with MAM attended every 2 weeks and received only a nutritional consultation, unless they required medical attention. Children who were found not to be gaining adequate weight or not progressing towards recovery (non-responders) were referred for medical assessment and/or inpatient care. Defaulters were traced by community health workers and referred to care if they were found to still be malnourished (though they were not re-enrolled in the study). All non-nutritional components of treatment such as systematic medications, frequency of follow-up, and home visits to prevent defaulting were identical between the combined and standard protocol arms. Full details of the combined and standard protocols were previously published [33] and are summarized in S2 Table.

## Outcomes

The primary outcome was recovery in the per-protocol analysis, defined as reaching a MUAC measurement of $\geq 12.5$ cm and no edema for 2 consecutive visits (weekly for SAM and biweekly for MAM). Secondary outcomes included cost-effectiveness, death (including post-defaulting deaths confirmed during defaulter tracing and follow-up phone calls when possible), non-response (16 weeks in treatment without achieving recovery), transfer (to either a stabilization center or a different facility), and defaulting (3 consecutive missed visits). The outcome categories recovery, death, non-response, defaulting, and transfer were mutually exclusive. To account for MAM children attending visits biweekly, we extended the non-response cutoff for all children to 17 weeks to evaluate the status of MAM children who would not have come for a visit at 16 weeks. Clinic staff determined whether children had met specified criteria in the protocol and discharged them from further follow-up. The criteria for each outcome (e.g., recovery being defined by 2 consecutive visits with a MUAC $\geq 12.5$ cm and no edema) were prespecified in the study protocol, and outcomes were assigned according to the strict definitions described in the study protocol. This process ensured that any discharge errors by clinic staff would be identified and accounted for during the analysis. An additional outcome category of early discharge was added during the analysis phase to account for children mistakenly discharged by the clinic staff before their outcome could be ascertained. The providers in each of the clinics were given on-site practical trainings in anthropometry and refresher trainings. Details of the procedures for taking anthropometric measurements are available in S1 Text. Research officers (registered nutritionists and supervisors for clinic staff) oversaw and repeated measurements to ensure agreement with clinic staff. Analyses of length of stay, average daily weight gain (g/kg/day), and average daily MUAC gain (mm/day) were conducted on children who achieved recovery. All outcomes were measured at the individual level. A formal data and safety monitoring board was not planned due to the expected short duration of recruitment. Instead, a trial safety committee comprising an independent chair and a statistician conducted a safety review at the midpoint of the trial. Outcomes and adverse events, including hospitalizations and deaths, were reported to the safety committee at the midpoint of recruitment. The committee, upon reviewing the safety report, recommended continuation of recruitment with no amendments to the protocol [33].

To assess cost-effectiveness, we calculated costs from a societal perspective, using accounting data, interviews with key informants, and survey questionnaires given to a subset of staff and all caregivers. Details have been described in our cost analysis methods paper [34]. We categorized costs into treatment, outreach, supply logistics, supervision, management, and household costs for each country.

We used a modified version of the Semi-Quantitative Evaluation of Access and Coverage (SQUEAC) assessment to assess coverage [39]. The SQUEAC method uses Bayesian techniques with a small sample survey to produce single coverage estimates. The coverage surveys assessed only MUAC and edema, in keeping with the study definitions of SAM and MAM. Coverage estimates are presented for SAM (MUAC < 11.5 and/or edema) and MAM (MUAC 11.5 to <12.5 cm) in each country and for each arm.

## Statistical analysis

Our sample size was designed to detect non-inferiority of recovery for the combined protocol compared to the standard protocol in the per-protocol analysis with a 10% non-inferiority margin and 80% power at a 2-sided level of significance of 5%. Based on these parameters, 12 clusters per arm were required, with 100 children in each cluster. The 10% non-inferiority margin was selected based on previous program data reporting a recovery rate of 85%, with a

minimum acceptable recovery rate of 75% per Sphere standards [40]. An intra-cluster correlation coefficient of 0.05 was assumed, based on results of a similar trial [21]. An additional adjustment to the sample size calculation was made to anticipate losses to follow-up and crossovers (children who are cross-registered across arms and need to be excluded during the analysis), in order to determine how many children should be recruited per cluster to ensure 100 children with completed treatment for the per-protocol analysis. Thus, the sample size was calculated to recruit enough children per cluster so that when defaulters (and other non-adherents, such as transfers or early discharges) were removed from the denominator, the power of the study would be maintained. Losses to follow-up were expected to be 15%, based on Sphere standards, and crossovers were expected to be 5%, so we aimed to recruit 150 children per cluster to achieve our target of 100 per cluster for the per-protocol analysis [33,41,42].

We used CommCare for data management [43]. In Kenya, clinic-based research officers entered data on Samsung Galaxy Tab A tablets running Android OS v.5.1.1 and uploaded data to the CommCare server. In South Sudan, nutrition supervisors initially collected data on paper forms. Data were then entered by 2 data entry clerks into Samsung Galaxy Tab A tablets running Android OS v.4.4 and uploaded to the CommCare server. Data entry onto paper forms and tablets was supervised and cross-checked by field coordinators. Data entry errors were logged and checked by the field coordinators and the principal investigator before incorporation into the database [33].

We conducted both per-protocol and intention-to-treat analyses. We designed this trial with per-protocol as the main analysis because it is more conservative and is the recommended approach for primary analysis of non-inferiority trials [33,44,45]. Non-inferiority was assessed only for the primary outcome of recovery. For secondary outcomes we explored evidence of a difference, rather than non-inferiority. The intention-to-treat analysis included all children enrolled, and the per-protocol analysis excluded children who did not complete treatment: those who exited the study as transfers to other facilities, defaulters lost to follow-up, or early discharges from treatment. Non-inferiority of the primary outcome of recovery was defined as a difference in proportion of children recovering from acute malnutrition with the upper bound of the 95% confidence interval being less than 10%. We used binomial regression models for binary outcomes and linear regression models for continuous outcomes. In both approaches we used clustered robust standard errors to adjust for repeated measures (clustering) within health facilities. We adjusted for the baseline characteristics of age, sex, and country in all the adjusted analyses. For costs with the greatest uncertainty we conducted a univariate sensitivity analysis. We present a mean base-case value across both countries, in US dollars, for cost per child treated, cost per child recovered, and the incremental cost of implementing the combined protocol compared to standard treatment. Analysis was conducted using Stata/IC v.13.1 [46]. Stata's zscore06 package was used to calculate WHZ, height-for-age $z$-score, and weight-for-age $z$-score (according to WHO Child Growth Standards) and flags [47].

The trial is registered at ISRCTN: trial number ISRCTN30393230 (http://www.isrctn.com/ISRCTN30393230). This study is reported according to the Consolidated Standards of Reporting Trials (CONSORT) guidelines (S3 Text; S4 Text).

## Results

Recruitment began 8 May 2017 and ended 31 March 2018. Trial activities were completed in Kenya on 31 July 2018 and in South Sudan on 31 August 2018, when the last enrolled children completed treatment. Among the children attending nutrition services at the participating health clinics, 157 children had WHZ < −2 but did not meet eligibility criteria because they

had a MUAC $\geq$ 12.5 cm and no edema; these children were enrolled in a sub-study and their outcomes will be presented in a forthcoming paper. A total of 4,110 children met eligibility criteria and were enrolled (Fig 1). Of these, 30 were excluded from analysis due to a missing admission date. Two more were excluded due to implausible MUAC values. The intention-to-treat analysis included 24 clusters and 4,078 children. After removing those who did not complete treatment, the per-protocol analysis included 24 clusters and 2,488 children (Fig 1). The target per-protocol sample size of 100 children per cluster was met: In the combined protocol arm the median sample per cluster was 107.5 (IQR 99.5–118); in the standard protocol arm the median sample per cluster was 102.4 (IQR 97–107.5).

Baseline characteristics were similar in the 2 study arms (Table 1). There were more females than males in the sample, and children had a mean age of around 17 months. About three-quarters of the children were aged between 6 and 24 months. Baseline characteristics by country are shown in S3 Table. Inter-country differences include the following: children in South Sudan were on average older than in Kenya (mean 22 versus 12 months); the Sudanese population was more rural, with household income mainly from fishing and farming; the Kenyan setting was more urban, with most household income from shopkeeping, casual labor, and salaried work; maternal education level in South Sudan was low, whereas most mothers in Kenya reported secondary level education (S3 Table).

In the per-protocol analysis, children in the combined protocol arm had non-inferior recovery (76.3% in the combined arm versus 73.5% in the standard arm; adjusted risk difference 0.03, 95% CI −0.05 to 0.10, $p$ = 0.52) (Table 2; Fig 2). There was no evidence of a difference in the secondary outcomes death (1.8% versus 1.8%; adjusted risk difference 0.00, 95% CI −0.01 to 0.01, $p$ = 0.87) and non-response (21.9% versus 24.7%; adjusted risk difference −0.03, 95% CI −0.10 to 0.05, $p$ = 0.48). Results in the intention-to-treat analysis were similar, with non-inferior recovery in the combined protocol (47.6% in the combined arm versus 43.8% in the standard arm; adjusted risk difference 0.03, 95% CI −0.06 to 0.13, $p$ = 0.47) (Table 2; Fig 2). There was no evidence of a difference between arms in the risk of the secondary outcomes death (1.1% versus 1.0%; adjusted risk difference 0.00, 95% CI −0.01 to 0.01, $p$ = 0.79), non-response (13.7% versus 14.7%; adjusted risk difference −0.01, 95% CI −0.05 to 0.04, $p$ = 0.73); defaulting (24.7% versus 30.7%; adjusted risk difference −0.06, 95% CI −0.15 to 0.03, $p$ = 0.20), and transfer to inpatient care (1.7% versus 1.1%; adjusted risk difference 0.01, 95% CI −0.01 to 0.02, $p$ = 0.42) (Table 2). The intra-cluster correlation coefficient for recovery was 0.05 (95% CI 0.01 to 0.08) for the per-protocol analysis and 0.06 (95% CI 0.02 to 0.09) for the intention-to-treat analysis.

Among children who recovered, there was no evidence of a difference between the combined and standard protocols in length of stay (65.4 versus 65.0 days; adjusted mean difference −0.55, 95% CI −5.75 to 4.65, $p$ = 0.83), average daily weight gain (1.9 versus 1.9 g/kg/day; adjusted mean difference 0.08, 95% CI −0.13 to 0.29, $p$ = 0.42), and average daily MUAC gain (0.2 versus 0.2 mm/day; adjusted mean difference −0.01, 95% CI −0.04 to 0.02, $p$ = 0.45) (Table 3).

Coverage results were similar for both protocols, with overlapping confidence intervals for all results. In Kenya, the coverage for SAM (MUAC < 11.5 cm and/or edema) was estimated at 54.9% (95% CI 41.2% to 68.0%) and 52.9% (95% CI 39.1% to 66.2%) for the combined and standard protocol arms, respectively. For MAM (MUAC 11.5 to <12.5 cm), coverage was 48.6% (95% CI 38.0% to 59.4%) and 47.3% (95% CI 37.1% to 58.0%) for the combined and standard protocol arms, respectively. In South Sudan, coverage for SAM was estimated at 45.9% (95% CI 32.5% to 59.9%) and 62.5% (95% CI 47.8% to 75%) for the combined and standard protocol arms, respectively. For MAM, coverage was 21.3% (95% CI 14.9% to 29.2%) and 26.3% (95% CI 19.2% to 35.1%) for the combined and standard protocol arms, respectively (S4 Table).

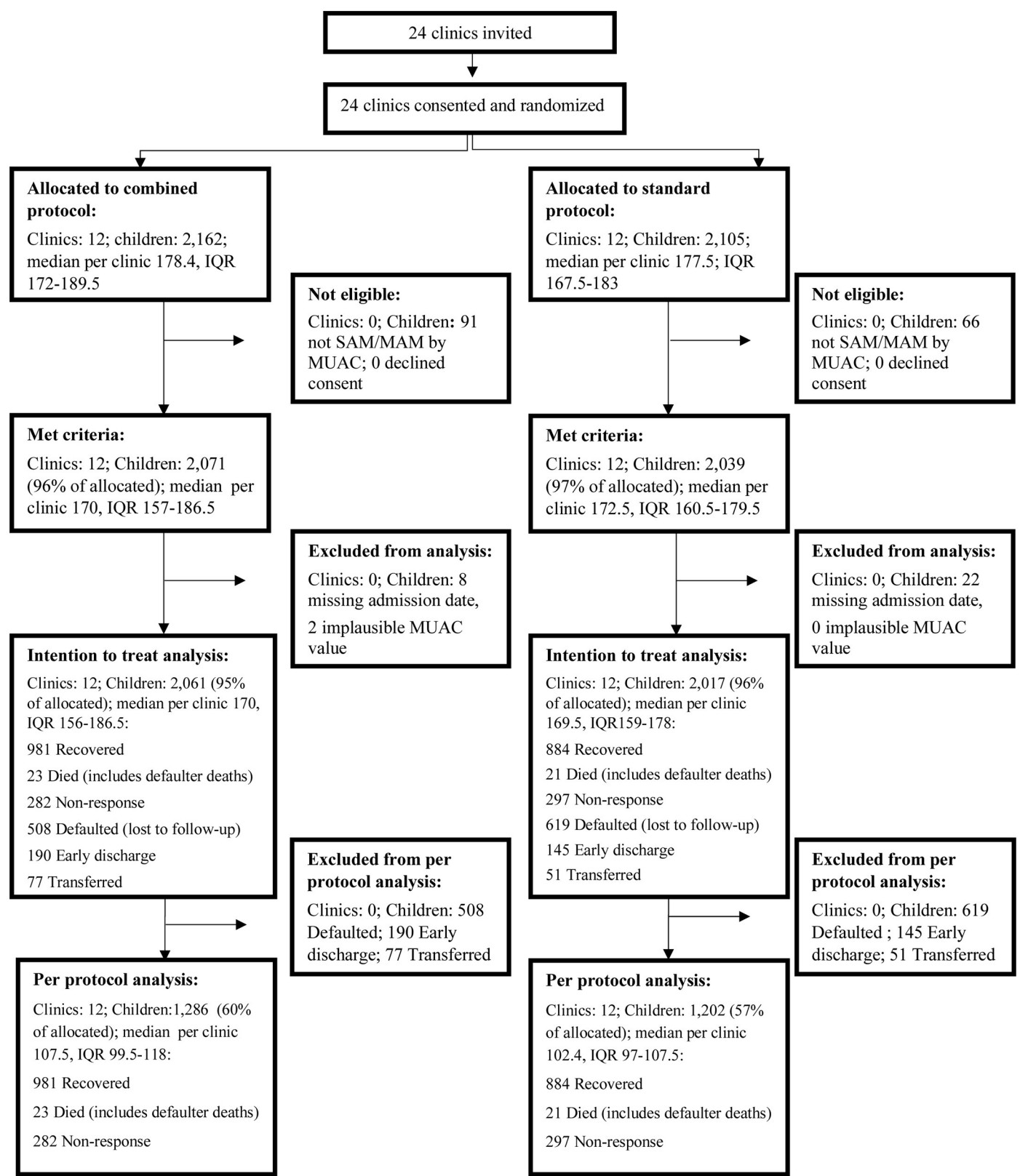

**Fig 1. Trial profile.** MAM, moderate acute malnutrition; MUAC, mid-upper arm circumference; SAM, severe acute malnutrition.

**Table 1. Admission characteristics of children in the combined and standard protocols.**

| Characteristic | Combined protocol (N = 12, n = 2,071) | Standard protocol (N = 12, n = 2,039) |
|---|---|---|
| **Sex and age** | | |
| Male, n (%) | 841 (41%) | 841 (41%) |
| Age in months, mean (SD) | 17.4 (11.4) | 16.7 (10.7) |
| Age category, n (%) | | |
| 6 to <24 months | 1,502 (73%) | 1,542 (76%) |
| ≥24 months | 564 (27%) | 495 (24%) |
| **Anthropometrics** | | |
| Weight (kg), mean (SD) | 7.2 (1.6) | 7.3 (1.6) |
| Height (cm)[a], mean (SD) | 72.9 (8.9) | 73.0 (8.7) |
| MUAC (cm), mean (SD) | 11.7 (.55) | 11.7 (.59) |
| WHZ, mean (SD) | −2.5 (1.0) | −2.3 (3.4) |
| HAZ, mean (SD) | −2.1 (1.6) | −1.9 (1.6) |
| WAZ, mean (SD) | −2.9 (1.0) | −2.7 (1.1) |
| MUAC < 11.5 cm, n (%) | 601 (29%) | 634 (31%) |
| WHZ < −3 | 352 (59%) | 346 (55%) |
| WHZ ≥ −3 to <−2 | 189 (32%) | 208 (33%) |
| WHZ ≥ −2 | 60 (10.0%) | 80 (12.6%) |
| MUAC 11.5 to <12.5 cm, n (%) | 1,461 (71%) | 1,397 (69%) |
| WHZ < −3 | 258 (18%) | 158 (11%) |
| WHZ ≥ −3 to <−2 | 650 (45%) | 550 (39%) |
| WHZ ≥ −2 | 553 (38%) | 689 (49%) |
| Edema (+ or ++), n (%) | 23 (1%) | 24 (1%) |
| **Other participant characteristics** | | |
| Child breastfed in last 24 hours, n (%) | 1,437 (69%) | 1,426 (70%) |
| Caretaker reports any morbidity[b] in child in past week, n (%) | 1,320 (64%) | 1,179 (58%) |
| Fever | 787 (38%) | 714 (35%) |
| Diarrhea | 521 (25%) | 517 (25%) |
| Cough | 651 (31%) | 694 (34%) |
| Health care sought in prior week | 489 (24%) | 339 (20%) |
| HIV status, n (%) | | |
| Positive | 9 (0%) | 5 (0%) |
| Exposed (mother) | 19 (1%) | 17 (1%) |
| Disabled (physically or mentally) | 44 (2%) | 38 (2%) |
| Tuberculosis (+), n (%) | 6 (0%) | 3 (0%) |
| **Household characteristics** | | |
| Mother is caretaker, n (%) | 1,978 (96%) | 1,923 (94%) |
| Maternal educational achievement, n (%) | | |
| None | 927 (47%) | 837 (44%) |
| Pre-primary | 9 (1%) | 13 (1%) |
| Primary | 475 (24%) | 485 (25%) |
| Secondary | 566 (29%) | 585 (30%) |
| College tertiary | 0 (0%) | 1 (0%) |
| Number of children under 5 years in the home, mean (SD) | 1.5 (1.0) | 1.7 (0.8) |
| Access to toilet, n (%) | 1,314 (64%) | 1,342 (66%) |
| Water source, n (%) | | |

(*Continued*)

**Table 1.** (Continued)

| Characteristic | Combined protocol (*N* = 12, *n* = 2,071) | Standard protocol (*N* = 12, *n* = 2,039) |
|---|---|---|
| Household tap | 92 (4%) | 174 (9%) |
| Community tap/tap stand | 684 (33%) | 742 (36%) |
| Hand pump/borehole | 746 (36%) | 518 (25%) |
| Borehole (private) | 162 (8%) | 452 (22%) |
| Vendors | 227 (11%) | 77 (4%) |
| Open water | 152 (7%) | 67 (3%) |
| Livelihood/main source of income, *n* (%) | | |
| No income | 10 (1%) | 112 (6%) |
| Sale of items (grass, firewood, livestock) | 324 (16%) | 97 (5%) |
| Fishing/farming | 649 (31%) | 559 (27%) |
| Business/shopkeeper | 226 (11%) | 586 (29%) |
| Casual labor | 529 (26%) | 586 (29%) |
| Salaried work | 301 (15%) | 284 (14%) |
| Other | 21 (1%) | 29 (1%) |
| Ever no food to eat[c], *n* (%) | | |
| Never | 1,022 (49%) | 1,294 (64%) |
| Rarely | 550 (27%) | 380 (19%) |
| Sometimes | 407 (20%) | 309 (15%) |
| Often | 73 (4%) | 43 (2%) |
| Don't know | 2 (0%) | 7 (0%) |
| Ever go to sleep without enough food[c], *n* (%) | | |
| Never | 1,205 (50%) | 1,189 (58%) |
| Rarely | 543 (26%) | 490 (24%) |
| Sometimes | 454 (22%) | 291 (14%) |
| Often | 39 (2%) | 43 (2%) |
| Don't know | 2 (0%) | 20 (1%) |
| Any household member go whole day without eating anything[c], *n* (%) | | |
| Never | 1,097 (53%) | 1,238 (61%) |
| Rarely | 507 (25%) | 438 (22%) |
| Sometimes | 421 (20%) | 279 (14%) |
| Often | 37 (2%) | 43 (2%) |
| Don't know | 1 (0%) | 35 (2%) |

*N* = number of clusters; *n* = individual children eligible for treatment.

[a]Recumbent length was taken for children 6–23 months, and standing height was taken for children 24 months and older.

[b]"Any morbidity" includes diarrhea, fever, cough, vomiting, skin infection, and other illness.

[c]From Household Hunger Scale.

HAZ, height-for-age *z*-score; MUAC, mid-upper arm circumference; WAZ, weight-for-age *z*-score; WHZ, weight-for-height *z*-score.

Kaplan–Meier plots showing the time to recovery are presented in S1–S6 Figs. Median time to recovery in the per-protocol analysis was approximately 10 weeks, with similar time to recovery in both arms.

Data from research grant accounts, NGO operational accounts, 83 key informant interviews, and 64 caregiver interviews were used to compute the total economic costs per

**Table 2. Primary and secondary outcomes (per-protocol and intention-to-treat analyses).**

| Outcome | Standard protocol | | Combined protocol | | Unadjusted | | Adjusted | |
|---|---|---|---|---|---|---|---|---|
| | n | Percent | n | Percent | Risk difference§ (95% CI) | p-Value | Risk difference§ (95% CI) | p-Value |
| *Per-protocol*[a] | | | | | | | | |
| **Primary outcome** | | | | | | | | |
| Recovery | 884 | 73.5% | 981 | 76.3% | 0.03 (−0.06 to 0.11) | 0.52 | 0.03 (−0.05 −0.10) | 0.52 |
| **Secondary outcomes** | | | | | | | | |
| Death | 21 | 1.8% | 23 | 1.8% | 0.00 (−0.01 to 0.01) | 0.95 | 0.00 (−0.01 to 0.01) | 0.87 |
| Non-response | 297 | 24.7% | 282 | 21.9% | −0.03 (−0.11 to 0.05) | 0.49 | −0.03 (−0.10 to 0.05) | 0.48 |
| *Intention-to-treat*[b] | | | | | | | | |
| **Primary outcome** | | | | | | | | |
| Recovery | 884 | 43.8% | 981 | 47.6% | 0.04 (−0.06 to 0.14) | 0.46 | 0.03 (−0.06 to 0.13) | 0.47 |
| **Secondary outcomes** | | | | | | | | |
| Death | 21 | 1.0% | 23 | 1.1% | 0.00 (−0.01 to 0.01) | 0.84 | 0.00 (−0.01 to 0.01) | 0.79 |
| Non-response | 297 | 14.7% | 282 | 13.7% | −0.01 (−0.06 to 0.04) | 0.66 | −0.01 (−0.05 to 0.04) | 0.73 |
| Defaulting | 619 | 30.7% | 508 | 24.7% | −0.06 (−0.15 to 0.03) | 0.20 | −0.06 (−0.15 to 0.03) | 0.20 |
| Transfer | 51 | 2.5% | 77 | 3.7% | 0.01 (−0.01 to 0.04) | 0.38 | 0.01 (−0.01 to 0.03) | 0.17 |
| Inpatient | 23 | 1.1% | 35 | 1.7% | 0.01 (−0.01 to 0.02) | 0.51 | 0.01 (−0.01 to 0.02) | 0.42 |
| New facility | 28 | 1.4% | 42 | 2.0% | 0.01 (−0.01 to 0.02) | 0.44 | 0.01 (−0.01 to 0.02) | 0.30 |
| Early discharge | 145 | 7.2% | 190 | 9.2% | 0.02 (−0.04 to 0.08) | 0.53 | 0.02 (−0.02 to 0.06) | 0.36 |

Unadjusted: all children with outcome measures, not adjusted for any demographic or study design characteristics. Adjusted: adjusted for age, sex, and country. $N$ = number of clusters; $n$ = number of children eligible for follow-up.

§Standard errors adjusted for clustering within facilities.

[a]Standard protocol: $N$ = 12, $n$ = 1,202; combined protocol: $N$ = 12, $n$ = 1,286.

[b]Standard protocol: $N$ = 12, $n$ = 2,017; combined protocol: $N$ = 12, $n$ = 2,061.

treatment arm and country (Table 4). To reach recovery in Kenya, SAM children in the standard protocol arm received on average 148 sachets of ready-to-use food (RUTF and RUSF), and SAM children in the combined protocol arm received on average 105 sachets of RUTF. To reach recovery in South Sudan, SAM children in the standard protocol arm received on average 209 sachets of RUTF and RUSF, and SAM children in the combined protocol arm received on average 133 sachets of RUTF. Overall, the average amount of ready-to-use food required for a child with SAM to reach full recovery (MUAC $\geq$ 12.5 cm and no edema for 2 consecutive visits) was lower in the combined protocol arm than in the standard protocol arm (122 versus 193 sachets). Combining mean costs and recovery rates across countries, per the statistical power of the study, the combined protocol costs US$123 less per child recovered than the standard protocol (Table 5). The input costs partially include the added costs of running a research trial, as these could not be fully separated from treatment program costs.

Among the 1,127 children who defaulted, 1,103 (97.9%) were followed up with home visits. Of these, only 386 (35.0%) were home at the time of the defaulter tracing visit and had their status verified. Among children traced with status verified, 157 (69.2%) in the standard protocol arm and 83 (52.2%) in the combined protocol arm were found to be still malnourished (with a MUAC < 12.5 cm and/or edema), 63 (27.8%) in the standard protocol arm and 72 (45.3%) in the combined protocol arm had recovered, 6 (2.6%) in the standard protocol arm and 2 (1.3%) in the combined protocol arm had died, and 1 (0.4%) in the standard protocol arm and 2 (1.3%) in the combined protocol arm were unknown. Among children with SAM, 284/641 (44.3%) defaulted in the standard protocol arm, and 209/617 (33.8%) defaulted in the

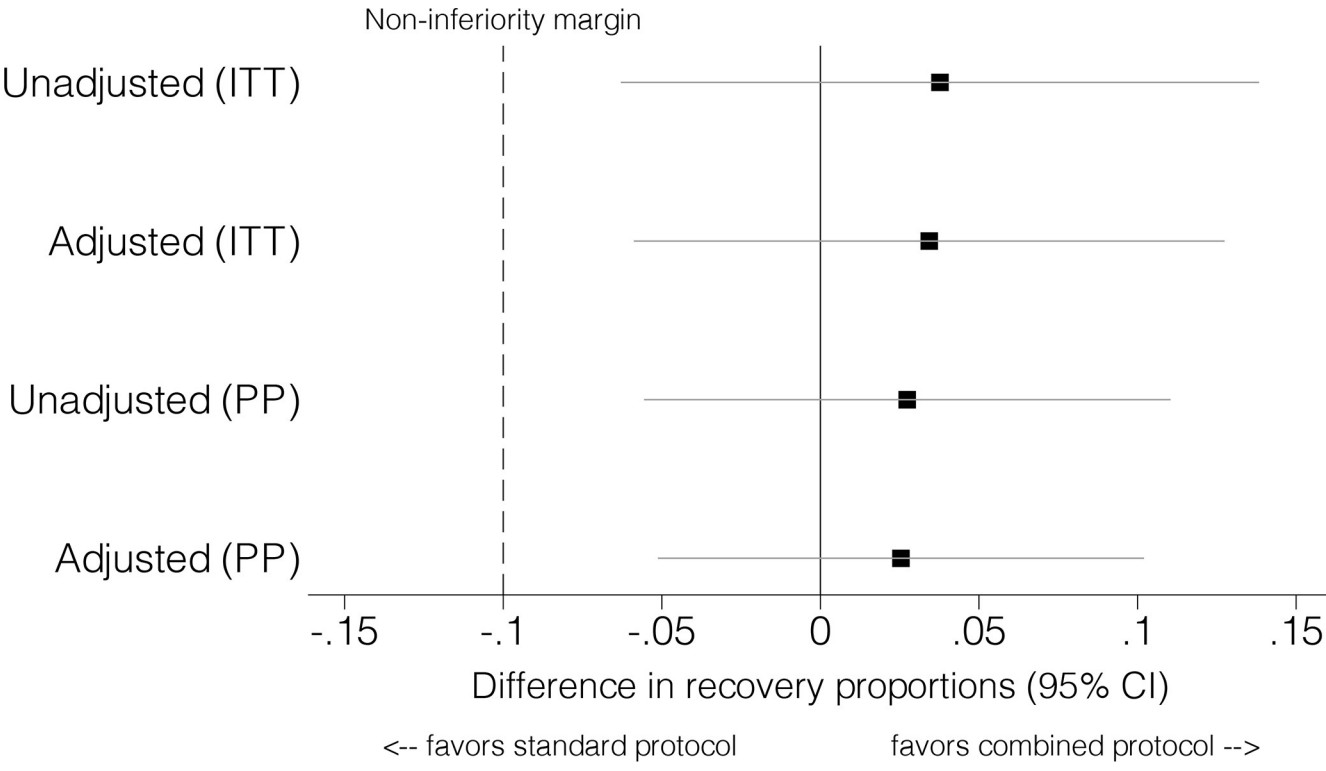

**Fig 2. Per-protocol (PP) and intention-to-treat (ITT) analyses of recovery.**

combined protocol arm. Among children with MAM, 348/1,376 (25.3%) defaulted in the standard protocol arm and 308/1,444 (21.3%) defaulted in the combined protocol arm.

In exploratory analyses of sub-groups, children with SAM, children with MAM, children aged <24 months, children aged ≥24 months, SAM children weighing ≥8 kg, children with a MUAC 11.5 to <12.5 cm but WHZ < −3, children in Kenya, and children in South Sudan recovered similarly under the combined protocol compared to the standard protocol (S5 Table).

## Discussion

ComPAS compared a simplified, combined protocol for acute malnutrition to standard care, and found non-inferior nutritional recovery in the combined protocol arm relative to the

**Table 3. Secondary outcomes among recovered children.**

| Outcome | Standard protocol (N = 12, n = 884) | | Combined protocol (N = 12, n = 981) | | Unadjusted | | Adjusted | |
|---|---|---|---|---|---|---|---|---|
| | Mean | SE | Mean | SE | Mean difference | p-Value | Mean difference | p-Value |
| Length of stay (days) | 65.0 | 2.31 | 65.4 | 2.24 | 0.42 (−6.19 to 7.04) | 0.90 | −0.55 (−5.75 to 4.65) | 0.83 |
| Average daily weight gain (g/kg/day) | 1.9 | 0.08 | 1.9 | 0.08 | 0.06 (−0.17 to 0.29) | 0.61 | 0.08 (−0.13 to 0.29) | 0.42 |
| Average daily MUAC gain (mm/day) | 0.2 | 0.01 | 0.2 | 0.01 | −0.01 (−0.05 to 0.02) | 0.39 | −0.01 (−0.04 to 0.02) | 0.45 |

Unadjusted: all children with outcome measures, not adjusted for any demographic or study design characteristics. Adjusted: adjusted for age, sex, and country. N = number of clusters; n = number of children eligible for follow-up.

MUAC, mid-upper arm circumference.

**Table 4. Input costs by country setting and treatment arm.**

| Cost category | Input costs—South Sudan | | Input costs—Kenya | |
|---|---|---|---|---|
| | **Combined protocol** | **Standard protocol** | **Combined protocol** | **Standard protocol** |
| Treatment (includes product) | $237,887 | $243,991 | $176,251 | $174,164 |
| Product only | $34,863 | $40,967 | $22,897 | $20,196 |
| SAM | $15,132 | $26,492 | $6,576 | $6,678 |
| MAM | $19,731 | $14,474 | $16,321 | $13,518 |
| Outreach | $71,329 | $71,329 | $20,766 | $20,766 |
| Supply logistics | $57,640 | $69,220 | $30,756 | $30,756 |
| Supervision | $128,917 | $128,917 | $14,125 | $15,919 |
| Management | $101,346 | $101,346 | $41,397 | $41,397 |
| Household | $8,861 | $9,891 | $11,563 | $11,680 |
| **Total** | **$605,979** | **$624,692** | **$294,859** | **$294,683** |

All values in US dollars.

MAM, moderate acute malnutrition; SAM, severe acute malnutrition.

standard care arm. The findings of the per-protocol and intention-to-treat analyses were similar. Clinical effectiveness was similar, with no evidence of a difference for the secondary outcomes death, non-response, defaulting, transfer to inpatient care, length of stay, program coverage, average daily weight gain, and average daily MUAC gain. Outcomes were similar between the study arms in exploratory analyses of sub-groups, including when disaggregated by SAM or MAM, and by country. The combined protocol cost less per child recovered than the standard protocol, especially for treatment of children with SAM and in the South Sudan context.

This study contributes evidence on integrated SAM/MAM treatment protocols, MUAC-based programming, and novel dosage regimes, which carry certain programmatic benefits: simplicity, ease of use, and potential cost-effectiveness. The combined protocol tested in this trial was the first to our knowledge to use a simple, MUAC-based dosage. Our findings are consistent with previous studies on integrated management of SAM and MAM with novel dosage regimes: a clinical trial in Sierra Leone that used a weight-based dosage protocol, with similar recovery and lower therapeutic food costs in the intervention arm of the study [21], and an operational trial in Burkina Faso that achieved recovery above Sphere standards and used less therapeutic food than standard programs [22].

**Table 5. Base-case cost-effectiveness results of the combined versus standard protocol.**

| Outcome | Combined protocol | Standard protocol |
|---|---|---|
| Total cost (US dollars) | $900,838 | $919,376 |
| Number of children in program | 2,071 | 2,039 |
| Recovery rate (intention-to-treat) | 48% | 44% |
| Number of children recovered | 981 | 884 |
| Cost per child treated (US dollars) | $435 | $451 |
| Cost per child recovered (US dollars) | $918 | $1,041 |
| Incremental cost (total) (US dollars) | −$18,538 | Ref |
| Incremental cost (per child recovered) (US dollars) | −$123 | Ref |

ITT, intention-to-treat; USD, US dollars.

Strengths of this study included comparison against the "gold standard" of standard care: community-based management of acute malnutrition at health clinics offering OTP and SFP in the same physical location, with RUSF offered to children with MAM instead of fortified blended flours. ComPAS was designed this way to capture the value added by simplifying and combining treatment. The reality is that many standard care programs are not able to provide RUSF for children with MAM, or the treatments for SAM and MAM are offered in separate physical locations. Another strength was the inclusion of 2 different contexts contributing to a combined dataset. The multi-country nature of the trial increases the generalizability of the results, with both rural and urban settings contributing data.

ComPAS faced several operational limitations. The contexts in Nairobi and Aweil East are very different, but the study sites were not large enough to be independently powered. Therefore, the analysis is only powered for combined multi-country estimates. Though our inclusion criteria included children up to 59 months, the majority were less than 24 months, so care should be taken in extrapolating the results to older or larger children. A non-inferiority margin of 10% was selected in consideration of previous program data from Aweil East and Nairobi. Actual proportions recovered are within a slimmer margin, with adjusted and unadjusted risk differences for both the per-protocol and intention-to-treat analyses within a 5%–6% margin (Table 2; Fig 2). In Aweil East, access to 4 of the clinics during the rainy season (May to November) was limited. Two of these clinics implemented the combined protocol, and 2 of the clinics implemented the standard protocol. One of the clinics was so inaccessible that supplies could not be delivered for more than 3 months. In Nairobi, mothers were often unable to take time off work to bring their children for regular visits, and some children with medical complications were not admitted to inpatient care for the same reason. The health clinics in Kenya were affected by a national nurse's strike from June to November 2017, which resulted in brief clinic closures and reduced availability of nursing staff. A national presidential election in August 2017, followed by a repeat election in October 2017, contributed to an atmosphere of uncertainty and some instability in the areas of Nairobi where the study took place. Many children were not able to attend their scheduled visits for several weeks during each election period, while families traveled to their rural homes to vote. All the factors above contributed to high defaulting in both contexts, reflected similarly in both arms, and may have negatively affected average daily weight gains. We have no reason to suspect that treatment acceptability had any role in defaulting. High defaulting drove down the overall recovery rates in the intention-to-treat analyses, which did not meet Sphere standards [40]. High defaulting also inflated the cost per child recovered. Defaulting did not impact our assessment of non-inferiority of recovery as the required number of participants completing treatment was met across clusters and arms.

The same underlying factors that drove high defaulting in both arms also contributed to frequent missed visits and longer lengths of stay. We do not attribute the longer lengths of stay to a difference in dosage protocol, as the lengths of stay were similar between arms. Where defaulting and missed visits are common, extending the period in treatment allows for more children to reach recovery. In our analyses, many children went on to recover after the 16-week cutoff for non-response (S1 Fig; S2 Fig).

The cost savings assessed in this trial were limited in scope. Though our method for assessing cost-effectiveness removed many of the research-specific costs [34], it was not possible to fully separate research versus programming costs in the analysis, so the total cost per child is higher than in many program settings. The costs reflect the combined inputs of NGOs and ministries of health, both of which contributed to the treatment of children. Additionally, the settings of Nairobi, Kenya, and Aweil East, South Sudan, have unique cost implications that may not affect other programs, such as the high costs of rent in a capital city and the high

logistics and security costs of operating a clinical trial in a conflict-affected context. Average costs in a non-research setting, although highly variable by context, are often around US$200 per child recovered [48,49].

There were large differences in costs between countries. It is important to note that costing results reflect the caseloads and characteristics of the sample at the country level, but the study is not statistically powered to make conclusions about differences in effectiveness. In South Sudan, there were 3 main reasons for cost savings in the combined protocol: (1) higher recovery and lower length of stay in the combined protocol arm (S5 Table), (2) a higher burden of SAM (with a reduced dosage of RUTF compared to the standard protocol), and (3) an independent supply chain. In Kenya, the larger proportional burden of MAM to SAM cases (S3 Table) and the presence of a centralized medical supply chain through the national government meant cost savings were not apparent. RUTF had a slightly higher price than RUSF (US $0.33 versus US$0.29 per sachet), which affected the cost of MAM treatment in the combined protocol.

Future studies should assess whether additional savings can be achieved from increased efficiencies throughout the larger system for managing acute malnutrition in operational settings. Further research is needed to assess cost-effectiveness in a scaled-up system treating SAM and MAM in 1 protocol with 1 food product across a variety of contexts. Operational studies could assess the impact of a simplified, combined protocol on SAM/MAM caseloads and resource use across multiple years. Potential cost savings may be possible through the prevention of SAM, with its associated medical complications and systematic medications. In the longer term, cost savings from increased efficiency and a reduction in SAM caseload may spread further upstream, addressing more of the MAM caseload. Earlier treatment of acute malnutrition is likely to result in fewer long-term implications, helping children to better survive and thrive.

Adoption of a simplified and combined protocol may simplify operations in resource-constrained settings. A combined protocol reduces the need to procure 2 separate products for SAM and MAM treatment, streamlines program logistics and staff training, and enables a more holistic continuum of care for children with acute malnutrition. A simplified protocol may also facilitate delivery of treatment by community health workers, to improve access and reduce defaulting, an important area of work that calls for research. In contexts where children are highly vulnerable to malnutrition—particularly in crises and chronically fragile situations —a combined protocol may offer some protection, enabling children to benefit from treatment before they deteriorate into life-threatening SAM. The optimal diagnostic criteria and dosage of therapeutic foods may differ by setting, and further research should explore these setting-specific issues and the operational and physiological impacts of expanding treatment of MAM [17]. The potential benefits of a simplified protocol may be weighed against economic considerations including implications for the supply chain of ready-to-use foods, and the impact on caseload and cost when treatment is made more readily available to moderately malnourished children. Further research is also needed to assess the safety and effectiveness of a reduced dosage protocol in children with SAM and in older or larger children. Though our study did not find evidence of harm among any sub-groups, including those with SAM and older children, future studies could be powered to assess those populations specifically. Our findings are consistent with other studies that indicate that a reduced dosage for SAM does not lower recovery [21,22,30,31], but research is needed to understand why a reduced dosage is effective and the contextual factors that affect recovery. Investigating the effects of context (e.g., differences between African and Asian settings, differences between food-insecure settings and food-secure settings, and the impact of seasonality) is a priority.

The reality is the current system fails to reach the majority of children who need treatment, and resources are already limited. If the global health community is to meet SDG nutrition

targets, new approaches are urgently needed. This is particularly true with the emergence of SARS-CoV-2, which has further necessitated a consideration of simplified, easy-to-use, and cost-effective nutrition treatment approaches. The number of malnourished children may rise as a result of the pandemic due to economic and biological factors, and the ability of the health system to respond may be strained. Simplified protocols can be incorporated into an adapted approach that could be delivered by community health workers at home and that optimizes the use of ready-to-use foods. Further research should assess how simplified nutrition protocols can be part of the response to this need, optimizing existing resources to reach more children while maintaining a high level of clinical care for all sub-groups. Streamlined, cost-effective programs make a more persuasive case for future investment, and this in turn is more likely to result in public health impact at scale.

## Supporting information

**S1 Fig. Time to recovery for all children in per-protocol analysis.**
(TIF)

**S2 Fig. Time to recovery for all children in intention-to-treat analysis.**
(TIF)

**S3 Fig. Time to recovery for SAM children in per-protocol analysis.**
(TIF)

**S4 Fig. Time to recovery for SAM children in intention-to-treat analysis.**
(TIF)

**S5 Fig. Time to recovery for MAM children in per-protocol analysis.**
(TIF)

**S6 Fig. Time to recovery for MAM children in intention-to-treat analysis.**
(TIF)

**S1 Table. Standard ready-to-use therapeutic food dosage table.**
(DOCX)

**S2 Table. Comparison of combined and standard protocols.**
(DOCX)

**S3 Table. Admission characteristics of children by country.**
(DOCX)

**S4 Table. Coverage survey results by country, arm, and SAM/MAM status.**
(DOCX)

**S5 Table. Analyses of recovery and length of stay by sub-group.**
(DOCX)

**S1 Text. Procedures for taking anthropometric measurements.**
(DOCX)

**S2 Text. ComPAS study protocol.**
(DOCX)

**S3 Text. CONSORT checklist for cluster trials.**
(DOCX)

**S4 Text. CONSORT checklist for non-inferiority trials.**
(DOCX)

## Acknowledgments

We thank the patients and their families for agreeing to participate in this study. We acknowledge the staff at each clinic, the ministries of health in Kenya and South Sudan, UNICEF and World Food Programme in Kenya and South Sudan, the London School of Hygiene & Tropical Medicine, the International Rescue Committee, Action Against Hunger, and the No Wasted Lives Coalition. Special thanks to Grace Heymsfield and Kayla Sossin.

## Author Contributions

**Conceptualization:** Jeanette Bailey, Charles Opondo, Bethany Marron, Mark Manary, André Briend, Marko Kerac.

**Data curation:** Jeanette Bailey, Charles Opondo, Natasha Lelijveld, Bethany Marron, Pamela Onyo, Eunice N. Musyoki, Susan W. Adongo, Marko Kerac.

**Formal analysis:** Jeanette Bailey, Charles Opondo, Natasha Lelijveld.

**Funding acquisition:** Jeanette Bailey.

**Investigation:** Jeanette Bailey, Charles Opondo, Natasha Lelijveld, Bethany Marron, Pamela Onyo, Eunice N. Musyoki, Susan W. Adongo, Mark Manary, André Briend, Marko Kerac.

**Methodology:** Jeanette Bailey, Charles Opondo, Natasha Lelijveld, Mark Manary, André Briend, Marko Kerac.

**Project administration:** Jeanette Bailey, Bethany Marron, Pamela Onyo, Eunice N. Musyoki, Susan W. Adongo.

**Resources:** Jeanette Bailey, Bethany Marron, Pamela Onyo, Eunice N. Musyoki, Susan W. Adongo.

**Software:** Jeanette Bailey, Charles Opondo, Natasha Lelijveld, Bethany Marron, Pamela Onyo, Marko Kerac.

**Supervision:** Jeanette Bailey, Charles Opondo, Bethany Marron, Pamela Onyo, Eunice N. Musyoki, Susan W. Adongo, Marko Kerac.

**Validation:** Jeanette Bailey, Charles Opondo, Natasha Lelijveld, Bethany Marron, Pamela Onyo, Eunice N. Musyoki, Susan W. Adongo, Mark Manary, Marko Kerac.

**Visualization:** Jeanette Bailey, Charles Opondo, Natasha Lelijveld, Marko Kerac.

**Writing – original draft:** Jeanette Bailey, Charles Opondo, Natasha Lelijveld, Bethany Marron, Pamela Onyo, Eunice N. Musyoki, Susan W. Adongo, Mark Manary, André Briend, Marko Kerac.

**Writing – review & editing:** Jeanette Bailey, Charles Opondo, Natasha Lelijveld, Bethany Marron, Pamela Onyo, Eunice N. Musyoki, Susan W. Adongo, Mark Manary, André Briend, Marko Kerac.

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
