## [Decision Letter · Decision Letter 0]

12 Mar 2020

Dear Dr. Bailey,

Thank you very much for submitting your manuscript "A simplified, combined protocol versus standard treatment for acute malnutrition (ComPAS trial): a cluster randomized controlled non-inferiority trial" (PMEDICINE-D-19-03592) for consideration at PLOS Medicine. 

[LINK]

In light of these reviews, I am afraid that we will not be able to accept the manuscript for publication in the journal in its current form, but we would like to consider a revised version that addresses the reviewers' and editors' comments. Obviously we cannot make any decision about publication until we have seen the revised manuscript and your response, and we plan to seek re-review by one or more of the reviewers. 

We expect to receive your revised manuscript by Apr 02 2020 11:59PM. Please email us (plosmedicine@plos.org) if you have any questions or concerns.

We look forward to receiving your revised manuscript. 

Sincerely,

Caitlin Moyer, Ph.D.

Associate Editor 

PLOS Medicine

plosmedicine.org

Title – please provide the country setting and add in children aged 6-59 months

The decimal point in numbers looks to be incorrectly formatted. Please do not place centrally and unbold. 

In the abstract please state what the secondary outcomes are. 

Data – please provide a link now and note that an author cannot be a point of contact for data access requests. 

CONSORT checklist – please use sections and paragraphs instead of page numbers as these can change during revisions and formatting etc. 

Comments from the reviewers:

Reviewer #1: This is a well organized paper on a much awaited trial.

Concerns:

Recovery rate in both ITT and per-protocol. In the ITT it barely meets the Sphere standard for the combined protocol. In the per protocol- it is substandard. While the authors note the reasons for defaulting, it is exceptionally low. it also make me question the acceptability of the programs overall. 

The two contexts in the RCT are extremely different and this variation dies not come out in the summary statistics. While the study was powered to only produce a combined analysis, South Sudan is really quite different that then slums of Nairobi and the supplemental table 4 demonstrates those programming variations- for example length of stay. In terms of the cost effectiveness, I see this as significant limitation. However, the authors address the limitations in the discussion. 

Reviewer #2: Bailey et al present the results of a cluster randomized non-inferiority trial of a simplified, combine protocol for SAM and MAM compared to standard treatment. This is an important trial and addresses a subject of much discussion in the nutritional programming community. The authors describe this trail, which took place in Kenya and South Sudan with children enrolled in a total of 24 clinics. The authors find that the combined treatment for SAM and MAM is non-inferior to standard care concerning recovery. The authors also provide cost-effectiveness information. 

There is no doubt that addressing this question adds to the literature. That being said, the article requires some aspects to be more clearly addressed by the authors.

1) There is an extraordinarily high loss to follow up (defaulting). As the authors state, this occurs in both arms, but is alarming. As such, the results, it could be argued, a limited to statement such as "among children remaining in the program, the two protocols were non-inferior." The differences between the ITT and PP analyses are such that this warrants a much larger discussion by the authors. Although balanced between arms, this is far above what would be acceptable in a program and limits what can be drawn from this trial.

2) The randomization warrants further clarity. From what I understand, the 24 health centers were selected, and then randomized to one of two groups. Then, the statistician was blinded until the database was locked? This is all fine, but then the randomization is both neither complex and the study is open label. Shouldn't the analyses have been done under blind? Is that what the authors mean? 

3) Although not typical, and this is greatly debated, I'm not entirely convinced about the power of this study given that the underlying assumptions for the ITT were very different than expected. It would be useful for the authors to comment a bit more on the analyses, provide additional information on lost to follow-up by cluster, and discuss further issues concerning the analyses themselves. This is also the case for the rationale for a cluster randomized trial, which is certainly warranted here, but doesn't come out clearly in the manuscript. Further to this, I would be more inclined to present the paper a bit more formally (was this a DSMB or not? It's unclear why Table 1 results are commented on? More details is needed in terms of the analyses

Minor comments

* Some of the references are duplicated

* The introduction could be updated to provide more recent and comprehensive information

* Some of the definitions are unclear, median? Z-score?

* The cost-effectiveness analysis is interesting, but it seems a bit less attention is paid to it in terms of framing this in a context of so many lost-to-follow up.

Reviewer #3: The authors have reported the results of a recent, large non-inferiority cluster-RCT comparing the effectiveness of a combined protocol against standard care of ready to use therapeutic foods to treat sever and moderate acute malnutrition for 6-59 month old children in Kenya an South Sudan. They have found that the simplified combined protocol was non-inferior to standard protocol as defined by their primary outcomes of difference in proportion who experienced nutritional recovery. The results are convincing and important, which will likely have impact to change practice, given the large sample size, recruitment across both rural and urban settings, and the base case economic analyses show that the simplified protocol is also cheaper. 

The manuscript is well-written, methodological sound in terms of recruitment methods, randomisation and analysis. I have very little critique and commend the authors on an impressive trial in a challenging environment. The recency of the trial also highlights a very important topic area of global significant with a large number of children suffering from acute malnutrition, in particular areas of Africa, leading to a high rate of morbidity of mortality. A simplified approach to nutrition supplementation may make it easier for clinics to operationalise and implement this approach in diverse settings. Future research should explore implementation research of the approach. 

Overall I have very little critique of the paper. My only minor comments are:

Methods - description of the analysis: Generally, cluster RCT analyses are a bit more complex - outcomes at follow-up can be analysed either at aggregrate level cluster level of individual outcomes, with both cases adjusting for baseline covariates. If individual level outcomes are analysed at follow-up (which it seems like this study) - then the analysis should allow for clustering using a mixed regression or generalised estimation equations. see Donner A, Klar N. Design and analysis of cluster randomization trials in health research. Arnold, 2000. It would be helpful if the authors could describe in method the type of model and regression technique they utilised. Line 226 simply states "regression analyses" which is too generic here. 

Methods - covariate adjustment - I see that the analyses have been adjusted for clustering. It's not clear from the methods section whether baseline adjustments to differences in children anthropometry were considered and adjusted. Given the unit of randomisation of a Cluster-RCT in a group, analyses will often adjust for baseline given repeated nature of the outcomes and individual differences within groups, but if the authors did not they should rationalise why not (i.e. expected differences between groups were expected to be minimal at baseline). 

Methods - it would helpful to specify that the z-scores were based on the WHO reference standards for growth charts. 

Results - subgroup analyses: Given there are two geographical groups: rural and urban - was there any thought given to a potential sub-group analyses in urban and rural localities (i.e. most mothers in Aweil East report no formal education where as mothers in Nairobi reported secondary level attainment). 

Reviewer #4: Compas trial, Bailey et al. 

This is an important trial and ditto manuscript and the authors should be congratulated on the large piece work and efforts put in this. I was impressed by the numbers in both groups, the non-inferiority as well as the costs saved. 

Introduction: one of the major things that are missing in the current rationale and introduction is in lines 84-87, page 4: "Evidence indicates…recover (Refs) till "but a study….dosage". Since the other arguments to come up with a combined approach for MAM (without danger signs) and uncomplicated SAM make sense, the authors need to elaborate in the introduction what this "evidence" indicates. The 3 references mentioned (24, 25, 26) should be briefly discussed or put into context. Did the authors in the papers/studies that are referred to use (in-)direct calorimetry, metabolic demands (RQ) to estimate and measure energy expenditure and usage? Was this done in the pre- or post RUTF era and were the study participants MAM or SAM patients? Lastly, it is not clear at present 'why' one would want to have a reduced dosage (line 87). Since all these arguments are underlying the study rationale, this needs more explanation to better put this trial into perspective.

Page 5, Participants:

Line 120: it would be good to know the % of children that were enrolled via active case finding and the % that were enrolled without. Do the authors feel this might have been a source of bias?

P6, line 139: please clarify of the PI did all data-analyses and data interpretation?

Same page, line 145: the section on study procedures is quite difficult to read and comprehend and it took me 3 times to understand the work-flow. 

Based on what evidence are the recommendations regarding intake done ("Combined protocol clinics….to 1 sachet RUTF/day (line 149)".

Page 14: since this is underlying the main result, that there is no difference between the 2 protocols and hence non-inferiority; I would recommend stressing that there was no difference between the ITT analysis versus the PP analysis. 

Page 19, last few sentences of the results sections; the 'exploratory analyses' show that subgroups recovered similarly. This is (in addition to my previous comment) an important finding and can (should) be presented a bit more prominently than is currently done. 

The discussion reads well and is to-the-point. It would be interesting to elaborate a bit more on your ideas (lines 391-394) on the 'future research' you suggest. Although I think you are right here it would be good from an academic perspective to hear your thoughts were in the spectrum from MAM to complicated SAM the combined protocol might still work and were it might stop being effective (non-inferior) to have 1 combined protocol.

[LINK]

---

## [Decision Letter · Decision Letter 1]

27 May 2020

Dear Dr. Bailey,

Thank you very much for re-submitting your manuscript "A simplified, combined protocol versus standard treatment for acute malnutrition in children 6-59 months (ComPAS trial): a cluster randomized controlled non-inferiority trial in Kenya and South Sudan" (PMEDICINE-D-19-03592R1) for review by PLOS Medicine.

I have discussed the paper with my colleagues and the academic editor and it was also seen again by reviewers. I am pleased to say that provided the remaining editorial and production issues are dealt with we are planning to accept the paper for publication in the journal.

[LINK]

We look forward to receiving the revised manuscript by Jun 03 2020 11:59PM. 

Sincerely,

Clare Stone, PhD

Managing Editor 

PLOS Medicine

plosmedicine.org

Requests from Editors:

Please add the mean age (combined cohort or individual groups) around line 38, and breakdown by sex.

Please change to 0.03 at line 40, and we don't want 0.00 included at line 44

Please provide a call out to the CONSORT checklist in the methods section

 Please remove "TM" at line 180

Is there additional access info for ref 22?

Please update ref 24

Comments from Reviewers:

Reviewer #2: The authors have responded clearly to the comments and this revised version is much improved. All questions have been responded to and this is an important trial for the nutrition community. 

Reviewer #3: I have re-read the manuscript and the authors have addressed all the comments appropriately. I have no further comments to add. 

Reviewer #4: Dear Authors,

I am happy with the comments to my review. 

Kind regards, 

Wieger Voskuijl

[LINK]

---

## [Editor Report · Decision Letter 2]

9 Jun 2020

Dear Ms Bailey, 

On behalf of my colleagues and the academic editor, Dr. James Tumwine, I am delighted to inform you that your manuscript entitled "A simplified, combined protocol versus standard treatment for acute malnutrition in children 6-59 months (ComPAS trial): a cluster randomized controlled non-inferiority trial in Kenya and South Sudan" (PMEDICINE-D-19-03592R2) has been accepted for publication in PLOS Medicine. 

PRODUCTION PROCESS

PRESS

PROFILE INFORMATION

Thank you again for submitting the manuscript to PLOS Medicine. We look forward to publishing it. 

Best wishes, 

Clare Stone, PhD

Managing Editor 

PLOS Medicine

plosmedicine.org